# Bio-Inspired Systems in Nonsurgical Periodontal Therapy to Reduce Contaminated Aerosol during COVID-19: A Comprehensive and Bibliometric Review

**DOI:** 10.3390/jcm9123914

**Published:** 2020-12-02

**Authors:** Andrea Butera, Carolina Maiorani, Valentino Natoli, Ambra Bruni, Carmen Coscione, Gaia Magliano, Giulia Giacobbo, Alessia Morelli, Sara Moressa, Andrea Scribante

**Affiliations:** 1Unit of Dental Hygiene, Section of Dentistry, Department of Clinical, Surgical, Diagnostic and Pediatric Sciences, University of Pavia, 27100 Pavia, Italy; carolina.maiorani01@universitadipavia.it; 2DDS, Private Dental Practice, 72015 Fasano, Italy; valentinonatoliodn@gmail.com; 3Free Lancer, 03100 Frosinone, Italy; ambrabruni@libero.it; 4Free Lancer, 10121 Turin, Italy; carmen.coscionedh@gmail.com (C.C.); moressa.sara@gmail.com (S.M.); 5Free Lancer, 31100 Treviso, Italy; gaia.magliano@gmail.com; 6Free Lancer, 20019 Milan, Italy; giulia.giacobbo91@gmail.com; 7Free Lancer, 61121 Pesaro, Italy; morelli.alessia@hotmail.it; 8Unit of Orthodontics and Pediatric Dentistry, Section of Dentistry, Department of Clinical, Surgical, Diagnostic and Pediatric Sciences, University of Pavia, 27100 Pavia, Italy; andrea.scribante@unipv.it

**Keywords:** periodontitis, bio-inspired, nonsurgical periodontal therapy, dentistry, dental hygiene

## Abstract

Background: On 30 January 2020, a public health emergency of international concern was declared as a result of the new COVID-19 disease, caused by the SARS-CoV-2 virus. This virus is transmitted by air and, therefore, clinical practices with the production of contaminant aerosols are highly at risk. The purpose of this review was to assess the effectiveness of bio-inspired systems, as adjuvants to nonsurgical periodontal therapy, in order to formulate bio-inspired protocols aimed at restoring optimal condition, reducing bacteremia and aerosols generation. Methods: A comprehensive and bibliometric review of articles published in English. Research of clinical trials (RCTs) were included with participants with chronic or aggressive periodontal disease, that have compared benefits for nonsurgical periodontal therapy (NSPT). Results: Seventy-four articles have been included. For probing depth (PPD) there was a statically significant improvement in laser, probiotic, chlorhexidine groups, such as gain in clinical attachment level (CAL). Bleeding on probing (BOP) reduction was statistically significant only for probiotic and chlorhexidine groups. There were changes in microbiological and immunological parameters. Conclusions: The use of bio-inspired systems in nonsurgical periodontal treatment may be useful in reducing risk of bacteremia and aerosol generation, improving clinical, microbiological and immunological parameters, of fundamental importance in a context of global pandemic, where the reduction of bacterial load in aerosols becomes a pivotal point of clinical practice, but other clinical trials are necessary to achieve statistical validity.

## 1. Introduction

On 30 January 2020, a public health emergency of international importance was declared as a result of the new disease COVID-19, an infection caused by a virus never identified in humans, SARS-CoV-2. The virus belongs to the family Coronaviridae, genetically placed within the genus Betacoronavirus, with a distinct clade in the lineage B of the sub-genus Sarbecovirus as well as two non-human Sars-like strains. It is an RNA virus covered by a capsid and a peri-capsid, crossed by glycoproteic structures that give it the typical appearance of a corona; it binds to the cell thanks to the interaction of the spike protein with the cellular receptor angiotensin-convertin enzyme 2 (ACE2) [1,2,3]. In addition to the definition, the mode of transmission of the virus has also been configured, which occurs mainly through inhalation, ingestion and direct contact of the mucous membranes with droplets of saliva; it is also essential to remember that the virus can survive on hands, objects or surfaces that have been exposed to infected saliva [3].

In Italy, the reported cases have grown dramatically over time, leading the country to gain a prominent position in the international scenario of infected patients.

This emerging pandemic and its serious outbreak led the Italian government to promote drastic impact measures to flatten the infection curve and, in turn, prevent health systems from collapsing. In fact, the limitation of people moving away from home, the social distancing, the cessation of almost all work activities and the request to the population to use masks and protective gloves, have the aim of minimizing the likelihood of non-infected persons coming into contact with persons who have contracted the virus; some professions, however, have had to guarantee a public service, such as dentistry [4].

### 1.1. The Problem in Dentistry

On 15 March 2020, the New York Times published an article titled “The Workers Face the Greatest Coronavirus Risk”, where an impressive scheme has described that dentists, dental hygienists, secretaries, chair assistants and laboratory technicians are among the workers most at risk of being affected by COVID-19 [5]. So, on the one hand there is a request to continue to offer a service for emergencies, and on the other hand operators must be able to work safely, while being exposed to great risk: direct contact, and contact with biological liquids and aerosols. In this regard, an article has recently been published by researchers of the Wuhan University School and Hospital of Stomatology with the aim of providing recommendations to dentists, for the management of the patient: minimize operations that can produce droplets or aerosols and use low- or high-volume vacuum cleaners to reduce them [6]. Since the viral load in saliva is very high, rinsing with antiseptic mouthrinses can only reduce the infectious amount, but is not able to eliminate virus [7]. The Regional Federation of Medical Surgeons and Dentists has also made available guidelines for patient management, where the telephone triage is mandatory with simple questions (Figure 1), to be able to frame the patient and assess the existence of an impossible performance. In addition, it is necessary to provide adequate air exchange in the operating room after each individual patient and to have adequate personal protective equipment. Among the figures working in the dental team, one of the most exposed professionals is definitely the dental hygienist, both for the close working distance, and for the use of instrumentation able to produce a large amount of aerosol [8]. In fact, ultrasonic scalers produce contaminant aerosols [9], defined as the suspension of extremely fine particles in the air, which can be liquid, solid, or a combination of both, with a diameter of 50 μm or less [10], and which may transmit pathogenic micro-organisms dangerous to health [11,12]. It can therefore be said that most dental procedures produce droplets and contaminant aerosols. This risk is related to the number of pathogens present in the aerosol/spray, and some instruments, such as rotating instruments, ultrasonic scalers and piezo tools, produce a greater amount of spray and aerosols than other instruments such as air-to-water syringes. The SARS-CoV-2 virus has a high affinity for epithelial lung cells and those of the salivary glands (large number of ACE2 receptors), and during a oral hygiene session with an infected patient, a large amount of virus is excreted to the aerosol spray, which can be inhaled by the health professional; the ACE2 expressing cells in oral tissues might provide possible routes of entry for the 2019-nCov, and thus, the oral cavity might be a potential risk route of 2019-nCov infection [13,14]. Bizzocca et al. have attributed risk scores for the dental team and patients for each procedure to be: direct contact with saliva (score 1), direct contact with blood (score 2), production of low levels of spray/aerosol via air–water syringes (score 3), the production of high levels of spray/aerosol by use of rotating, ultrasound and piezoelectric tools (score 4): “tartar scaling” has one of the most high risk (7.5), such as some surgical or endodontic procedures [15].

Several strategies can be implemented to contain this problem, such as antiseptic pre-treatment rinsing, which helps control the infectious agents in aerosol [16,17], reducing the colonies by up to 94% [18], or the use of a rubber dam, which would seem to eliminate any contamination from blood and saliva. In fact, the latter has several limitations, as it is inapplicable in the procedures of scaling–root planning (SRP), periodontal surgery and prophylaxis routine. Reducing bacteriemia must be among the main objectives of nonsurgical periodontal therapy, regardless of the historical moment in which we are living: it is necessary to manage the production of contaminating aerosols, favoring therapies that make use of new technologies, which are increasingly minimally invasive, such as lasers, ozone and probiotics, and disinfectants with bactericidal action, such as chlorhexidine and ozone (although the latter require further studies to validate their effectiveness).

### 1.2. Oral Microbiota and Periodontal Disease

A well-balanced oral microbiota is essential in preventing the onset of oral cavity diseases. However, we do not always find ourselves in a state of eubiosis, and therefore in that physiological condition of functional balance in which bacteria and structures operate in a balanced manner, favoring absorption and regulation of the system. Very often, in fact, we move from a state of microbial equilibrium in which the oral microbiota produces the metabolites necessary for the human body, with positive effects for human health, and we move to a state of dysbiosis. In this particular condition, not only does the gene coding of useful molecules fail, but harmful compounds are partially metabolized by pathogenic microorganisms, also part of the microbiota. By the term of dysbiosis is meant, therefore, the pathological condition of functional imbalance of the digestive system, where the functions of the selective barrier are lost [19]. The clinical objective is to restore the symbiotic balance between bacteria and host.

Periodontal disease is a multifactorial, degenerative and irreversible disease affecting the tissues supporting the teeth: one of the main factors is undoubtedly the formation of bacterial biofilms. Its control, in fact, promotes the prevention and maintenance of good health by the oral cable [20].

Therefore, the basic periodontal treatment aims to eliminate hard and soft deposits above and below the gingiva, establishing the conditions to allow effective control of the bacterial plaque: in this sense, scaling and root planning through the use of manual and ultrasonic instrumentation, turns out to be the gold-standard procedure for debridement of root surfaces, demonstrating clinical and microbiological benefits [21]. However, traditional nonsurgical periodontal therapy has limitations: longer sessions, less comfort for the patient, excessive instrumentation (overtreatment); it is also performed in several sessions, usually one appointment for each quadrant, so there is the possibility of bacterial recolonization with consequent delay in the healing of the patient [22].

In the context of the increasing new approaches to periodontal treatment and in view of the increasing possibility of a proactive therapy, that is, a therapy able to solve clinical problems at the first symptoms or to prevent them altogether, bio-inspired protocols have been developed for the achievement of health conditions and the respect of tissues, favoring the restoration of a microbiological balance. This is made possible by the new technology available, which has enabled the introduction of different methods for the treatment of patients with periodontal disease, using the support of laser devices, ozone therapy, airflow powders based on glycine and/or erythritol, and also the use of probiotics, thus reducing the chemical–pharmacological action. From the literature, we know that the laser has been extensively studied to highlight its benefits in nonsurgical periodontal treatment. From different clinical trials, in fact, it is clear that the main effects are found in the reduction of the probing depth and in the gain of the clinical attachment with the use of a Er:YAG laser (as described in [23], after nonsurgical periodontal therapy in [24], and laser compared with SRP alone in [25]), but also Nd:YAG laser was used in cases where there was bleeding on probing, after nonsurgical periodontal therapy with ultrasonic and hand instruments [26,27]. Further improvements were found in the decrease in levels of IL-1β and TNF-α following photodynamic therapy [28]. The lasers used were used at different wavelengths, even surgically, associated with or compared to nonsurgical periodontal therapy. Additional benefits are evident with the use of ozone devices in the most significant periodontal clinical parameters, resulting in reduction of inflammatory parameters in a quarterly follow-up, such as PTX-3, IL-1β, Hs CRP [29], and bacterial count [30], especially of aggregatibacter actinomycentemcomintans (reduction of 25%) [31]. As for glycine and/or erythritol powders, on the other hand, they are more frequently used in periodontal support therapy for the removal of the subgingival biofilm, with greater comfort for the patient [32,33,34]. Finally, the use of probiotics showed an improvement in the epidemiological indices of reference for periodontal disease [35], as well as a reduction of the bacteria belonging to the red and orange complex of socransky and the proinflammatory cytokines [36].

Additionally, it is known from the literature that chlorhexidine is the antibacterial compound with bactericidal action most used as a support of periodontal therapy, in favor of the reduction of the pocket depth [37,38] and in the change of the subgingival microbiota [39,40]; is mainly used in the form associated with the Xanthan gel, when the latter is able to stabilize the molecule in subgingival tissues for a sufficient time [38]. On the basis of this information and available knowledge, a study was conducted to verify the effects of chlorhexidine on the oral microbiome, and it emerged that it significantly increased the abundance of Firmicutes and Proteobacteria and reduced the content of Bacteroidetes, TM7, SR1 and Fusobacteria. This shift was associated with a significant decrease in saliva pH and buffering capacity, accompanied by increased levels of lactate and saliva glucose. Lower concentrations of saliva and plasma nitrites were found after the use of chlorhexidine, followed by a tendency to increase systolic pressure. Overall, this study shows that mouthwash containing chlorhexidine is associated with an important change in the salivary microbiome, which leads to more acidic conditions and a lower availability of nitrites [41]. In addition, it is shown that a rinse with chlorhexidine prior to any dental procedure can reduce the likelihood of cross-infection, due to the presence of bacteria in the environment and the spread of aerosols [42], and the risk of bacterial infections [43], which can be induced by simple oral hygiene maneuvers, but also by more complex procedures, such as scaling and root planning [44]. In fact, periodontal disease contributes to systemic blood flow and to the migration of microorganisms, and their products, throughout the body [45,46]: bacteria have developed mechanisms to invade and adapt in host cells, escaping the host’s immune response and releasing free toxins, causing transient bacteremia. This explains the relationship between oral and systemic conditions and the importance of pre-treatment rinses with antimicrobial action, from which it is reasonable to expect a positive effect on bacteremia.

In order to approach bio-inspired systems, even in this historical context, without giving up the tools now needed in daily clinical practice, it is essential to focus on the concept of bacteremia and generation of aerosols, to understand how to best manage the instrumentation, evaluating scientifically validated protocols in literature: all these protocols, aim at the reduction of bacteriemia and therefore the bacterial load present in aerosols.

A patient with altered periodontal status needs strict controls that aim at restoring biological conditions and are minimally invasive, such as the use of ultrasonic inserts that are best suited to the shape of the element taken into consideration, or diamond inserts, if forks are also involved; the aggressiveness of a manual instrumentation would lead to a the re-entry of bleeding and periodontal pockets, but also to a greater loss of tissue, that is, an increase of the gum recessions and probable loss of adherent gums. Moreover, in an environment that is increasingly being defined by the mini-invasiveness of any therapy that aims at restoring the aesthetic, also antimicrobial agents should be reviewed. Similar results to those obtainable with the irrigation of the pockets with chlorhexidine, are with the use of ozone, whether in liquid, gaseous or gel form for an immediate decontamination of the grooves and/or periodontal pockets, being a powerful natural disinfectant. Additionally, for the control of gingival inflammation and for a proper restoration of the oral microbiome, an alternative may be the use of lactobacilli-based probiotics, administering two of them daily, in such a way as to reduce the bacterial population, particularly streptococci. Another aspect to take into account during the treatment of periodontal disease is the treatment of hard tissues with bio-inspired materials. Biomimetics nanohydroxyapatite products can be recommended for the remineralization of dental enamel, allowing a more accurate retention of surfaces [47], especially during orthodontic treatment than can alter the oral environment [46] and detersion efficacy [48].

Bad oral hygiene habits can encourage the accumulation of periodontal pathogens in the oral cavity and dysbiosis can accelerate the decline of lung function: in addition, pathogenic bacteria such as treponema denticola, P. gingivalis, fusobacterium nucleatum, aggregatibacter actinomycetemcomitans and veillonella parvula, were found in the lungs of patients admitted to the ICU [49]. Their presence, can not only change the microbial composition of the respiratory system, but also promote a number of responses of cytokines, affecting the immune homeostasis of the lungs: spherical levels of IL-6 and IL-8 increase significantly in patients with pulmonary dysfunction and local inflammatory factors spread into the systemic circulation. Changes in cytokines are assumed to reflect the state of the disease to a certain extent [50].

A high bacterial and viral load in the mouth can lead to complications in systemic diseases such as cardiovascular diseases, neurodegenerative diseases and autoimmune diseases, further supporting the bond between the mouth and the body: risk factors established for COVID-19 (age, sex and comorbidity) are also strongly implicated in imbalances in the oral microbiome. In fact, diabetes, blood hypertension and heart disease are associated with a greater number of F. nucleatum, P. intermedia and P. gingivalis, favoring the progression of periodontal disease: Patients with periodontal disease increase the risk for cardiovascular disease by 25%, for high blood pressure by 20% and triple the risk for diabetes mellitus [51,52,53,54]. Epithelial sensitization and hematogenic diffusion of proinflammatory mediators such as cytokines, produced in the periodontal diseased tissue, can increase systemic inflammation and decrease airflow: this can be exacerbated by the stimulation of the liver to produce acute phase proteins, such as interleukin-6, which boost the inflammatory response of the lungs and the rest of the body. Similarly, patients with COVID-19 in severe form also express systemic inflammation and higher levels of IL-6, IL-2, IL-10, TNF and C-reactive protein [55].

## 2. Materials and Methods

### 2.1. Focus Question

Can the use of lasers, ozone, probiotics, glycine and/or erythritol, chlorhexidine in combination with nonsurgical periodontal treatment have additional beneficial effects on the clinical parameters of periodontal disease? Can these bio-inspired instruments be used to reduce the risk of bacteremia during COVID-19 disease?

### 2.2. Elegibility Criteria

First, we have analyzed studies in accordance with the following inclusion criteria:

Type of studies. Randomized controlled clinical trials, controlled clinical trials, prospective clinical trials, in vivo retrospective clinical trials with the approval of the Ethics Committee.

Types of participants. Participants with chronic and/or aggressive periodontal disease were considered. (1) Patients undergoing periodontal surgery or nonsurgical periodontal treatment within three months prior to the beginning of the clinical trials examined were excluded; (2) undergoing maintenance therapy or periodontal support, or (3) treatment of residual pockets, following periodontal or nonsurgical surgical therapy; (4) patients with concomitant systemic pathologies that could have affected the periodontal outcome were excluded.

Type of interventions. Clinical trials that have compared benefits for scaling and root planning in quadrant or full-mouth (SRP/FMD). The experimental group assisted by one or more laser treatments such as, diode lasers, Er:YAG laser, Nd:YAG laser, Er, Cr:YSGG laser, photobiomodulation (PBM), photodinamic therapy (PDT); ozone treatments such as, ozone gas, ozone water, ozone gel; treatments with probiotics such as Lactobacillus or Bifidobacterium; treatments with glycine airpolishing or periopolishing; treatments with erythritol airpolishing or periopolishing; chlorhexidine treatments such as, chlorhexidine mouthwash, gel, chip or varnish. One or more control groups administered a placebo or control treatment other than the experimental one.

Outcome type. Primary outcomes: plaque index (PI), blindind on probing (BOP), probing depth (PPD), and clinical attachment level (CAL). Other clinical parameters, where present, such as gingival index (GI), gingival recession (GR(REC)), gingival margin index (MGI), modified bleeding index (MBI), gingival bleeding index (GBI), sulcular bleeding index (SBI), full mouth plaque score (FMPS), visible plaque index (VPI), visible plaque index (API) and microbiological and immunological parameters have been considered.

We have included in the second phase only those studies that met all the inclusion criteria, that is to say, the analysis of the selected studies according to the exclusion criteria: (1) clinical studies where the authors have not reported at least one of the clinical parameters chosen as outcomes; (2) clinical studies where participants have undergone periodontal surgery or nonsurgical periodontal treatment during the 3 months prior to the beginning of the clinical trials examined; (3) clinical studies performed on participants with concomitant systemic pathologies that could have affected the periodontal outcome; (4) studies performed on participants in support/maintenance therapy; (5) clinical studies carried out on participants for the treatment of residual pockets, following periodontal surgical or nonsurgical therapy; (6) clinical studies where laser, ozone, probiotics, glycine, erythritol or chlorhexidine have not been used as a test group; (7) in vitro or animal clinical studies; (8) clinical trials carried out without the approval of the Ethics Committee.

The risk of bias was determined evaluating: adequate sequence generated (participants should be allocated to groups, using a true randomization sequence), allocation concealment (participants and investigators should not be able to predict allocation before participants are included in the study), blinding (participants and investigators should be unaware of the allocation to ensure that everyone gets the same amount of attention; the blindness of those who evaluate the outcomes may reduce the risk that knowledge of the intervention received, influence the measurement of outcomes), incomplete outcome data (report should describe how the outcomes have been measured in the method section) and registration of outcome (results should be reported for each outcome identified at the beginning).

### 2.3. Search Strategy

The review is based on the research of clinical trials (RCTs) in reference to the PICOT model (Population, Intervention, Comparison, Outcome, Timing), identified through bibliographic research in electronic databases, examining the bibliography of articles, on Pubmed (MEDLINE) and Google Scholar. Initially, all abstracts of clinical studies published from January 2010 to March 2020 were taken into consideration which evaluated the effect of the addition of laser therapy, ozone therapy, probiotics, glycine and erythritol, chlorhexidine to nonsurgical periodontal therapy in the treatment of periodontal disease.

### 2.4. Research

We performed the search using: “periodontal disease”, “periodontitis”, “nonsurgical periodontal treatment”, “laser”, “RCTs AND laser AND periodontal disease”, “laser AND periodontitis”, “laser AND nonsurgical periodontal treatment”, “ozone”, “RCTs AND ozone AND periodontal disease”, “ozone AND periodontitis”, “ozone AND nonsurgical periodontal treatment”, “probiotics”, “RCTs AND probiotics AND periodontal disease”, “probiotics AND periodontitis”, “probiotics AND nonsurgical periodontal treatment”, “RCTs AND glycine AND periodontal disease”, “glycine AND periodontitis”, “glycine AND nonsurgical periodontal treatment”, “erythritol”, “RCTs AND erythritol AND periodontal disease”, “erythritol AND periodontitis”, “erythritol AND nonsurgical periodontal treatment”, “chlorhexidine”, “RCTs AND chlorhexidine AND periodontal disease”, “chlorhexidine AND periodontitis”, “chlorhexidine AND nonsurgical periodontal treatment”. We have included patients with chronic periodontitis or aggressive periodontitis, based on the classification of periodontal diseases proposed by Armitage in 1999 and the new classification presented on 22 June 2018 on the occasion of the Europerio9.

### 2.5. Screening and Selection of Articles

Titles and abstracts were collected, in which were present the search keywords and the information related to the inclusion criteria to proceed with the reading in full-text. After reading in detail, the studies that met all the selection criteria were evaluated, to then extract and analyze the data collected.

### 2.6. Search Outcome and Evaluation

The first research outcomes were PI, BOP, PPD, CAL. Other interesting outcomes, where present, were the changes in the subgingival plate. Information was extracted from each study on (1) participants’ characteristics (age and disease characteristics) and criteria for inclusion and exclusion from the clinical trial in question; (2) intervention (modality) vs. placebo or vs. no treatment or vs. comparison treatment (different from the one tested, therefore different from laser, ozone, glycine, erythritol, probiotics and chlorhexidine); (3) outcome (possible improvement of the clinical parameters examined for the treatment groups included); (4) clinical data examined (PPD, CAL, BOP, PI); (5) other clinical data (possible microbiological evaluation); (6) follow-up.

## 3. Results

### 3.1. Study Selection

A total of 458 articles on the use of minimally invasive technologies in nonsurgical periodontal treatment emerged from several researchers. Subsequently, from a first reading of the abstracts found, we eliminated: (1) articles emerged in more researches carried out; (2) review and meta-analysis; (3) articles on the treatment of mucositis and peri-implantitis; (4) articles on the treatment of gingivitis; (5) articles on the treatment of gingivitis and periodontitis in orthodontic treatment. A total of 96 studies were therefore identified (52 laser studies; 8 ozone studies; 10 airpolishing studies; 25 probiotic studies; 21 chlorhexidine studies) on nonsurgical periodontal treatment, approved by the Ethics Committee.

In a second phase, following the full-text reading, 11 clinical studies (4 inherent to the laser; 1 inherent to the ozone; 1 inherent to the airpolishing; 4 inherent to the probiotics; 1 inherent to the chlorhexidine), of which there was only the availability of reading the abstract and further 10 studies not in compliance with the eligibility criteria.

74 articles have been included (36 RCTs nonsurgical periodontal therapy (NSPT) + laser; 5 RCTs NSPT + ozone; 3 RCTs NSPT + airpolishing; 15 RCTs NSPT + probiotics; 15 RCTs NSPT + chlorhexidine; Figure 2).

### 3.2. Characteristics of Studies

#### 3.2.1. NSPT and Laser

Methods. The 36 studies selected for the review were randomized clinical trials published in English. The duration of studies varied from 1 to 12 months for a total average of about 5 months (5.25), where 5.5% of studies had a follow-up of 1 month, 2.8% of 2 months, 30.6% of 3 months (these studies have some limits related to pocket depth and clinical attachment gain, because the follow-up is too short to allow the re-attachment), 50% of 6 months and 11.1% of 1 year. The 27.8% of the studies analyzed were conducted in Turkey, a further 27.8% in Brazil and India (equally distributed), 8.3% in Spain, 11.1% in Italy and China (equally distributed) and 25% in Germany, the Netherlands, Poland, Croatia, Greece, Iran, Serbia, Arabia and Thailand (equally distributed). Randomization of the studies was performed with different methods: computer-generated table, where the majority of allocation concealment was done through the use of opaque sealed envelopes, or tossing/flipping a coin; the 58.3% of clinical trials was designed in split-mouth.

Participants. Studies on average recruited about 31 patients (31.61), where 66.7% of studies included ≤31 patients and 33.3% of studies included >31 patients. The main inclusion criteria included: age ≥18 (100% of the studies examined), presence of at least 2 teeth with at least 1 site with PPD and bleeding on probing in each quadrant, have not undergone periodontal surgery or nonsurgical periodontal treatment within 3 months prior to the start of clinical trials, without any systemic pathology that could affect periodontal clinical parameters and patients who did not require antibiotic prophylaxis for dental treatments.

Interventions. Patients underwent nonsurgical periodontal treatment (scaling and root planning in full-mouth or in quadrants) in addition to laser therapy for the test group. The 25% of the studies used the PDT, 19.4% the diode laser, 16.7% Er:YAG laser, 16.7% used Nd:YAG laser or Er,Cr:YSGG laser (equally distributed), 13.9% the PBM, 5.6% used a combination with Er:YAG and Nd:YAG laser and 2.7% of the studies used, instead, a combination of PDT and PBM. The control group was only subjected to nonsurgical periodontal therapy. Some of the studies included used one or more test groups (eight clinical trials, 22.2% of the total studies) and one or more control groups (one clinical trial, 2.8% of the total studies). The patients have undergone scaling and root planning with ultrasonic and hand instruments: the laser was used, in most cases, immediately after nonsurgical periodontal treatment, except for some lasers (such as Er:YAG) that were used before therapy.

Primary outcomes. In most of the studies they were PPD and CAL; in one study, among the primary outcomes, was also the bacterial count; all studies have evaluated every type of adverse event. The frequency of evaluation of outcomes was variable: monthly, quarterly, half-yearly or a single final evaluation at one year.

Secondary and additional outcomes. BOP, PI (or VPI, FMPS), API GI, SBI, bleeding index (BI), GR (REC) were clinical parameters evaluated in addition to the probing depth and the loss of clinical attachment; they have not been recorded in all the studies examined. Some studies have also performed microbiological and immunological (61.1% of the total); other studies: one study evaluated the perception of the patient during treatment; one study evaluated fluctuation of periodontal somatosensory function and gingival microcirculation and two further studies evaluated the patient’s halitosis.

#### 3.2.2. NSPT and Ozone

Methods. The 5 studies selected for review were randomized clinical trials published in English. The duration of studies varied from 1 to 3 months for a total average of about 2 months (2.2), where 20% of studies had a follow-up of 1 month, 80% a follow-up of 2 and 3 months (40% each). Additionally, in these clinical trials the follow-up is too short for evaluate improvements in clinical parameters. The 60% of the studies were conducted in Turkey, while the remaining clinical trials were conducted in Japan and Poland. Randomization of studies was performed with different methods: computer-generated/randomization list or tossing a coin; the 40% of studies were designed in split-mouth.

Participants. Studies on average recruited about 32 patients (32.2), where 40% of studies included ≤32 patients, while 60% of studies included >32 patients. The main inclusion criteria included: age ≥18 (100% of the studies examined), presence of at least a 2 teeth with at least 1 site with PPD and bleeding on probing in each quadrant, have not undergone periodontal surgery or nonsurgical periodontal treatment within 3 months prior to the start of clinical trials, without any systemic pathology that could affect periodontal clinical parameters and patients who did not require antibiotic prophylaxis for dental treatments.

Interventions. Patients underwent nonsurgical periodontal treatment (scaling and root planning in full-mouth or in quadrants) in addition to ozone therapy. Most studies used ozone in gaseous form, while one study evaluated the effect of NBW3 (ozonated water nanobubbles). The patients have undergone to scaling and root planning with ultrasonic and hand instruments: ozone was used immediately after periodontal treatment.

Primary outcomes. In most of the studies they were PPD and CAL. In one study they were included as primary outcomes PPD, GI, PI and BOP (the difference between these variables was then used as secondary outcomes); all studies evaluated each type of adverse event. The frequency of evaluation of outcomes was variable: monthly or quarterly.

Secondary and additional outcomes. BOP, PI, GI, SBI, API are clinical parameters evaluated in addition to the probing depth and the loss of clinical attachment; they have not been recorded in all the studies examined. A microbiological and/or immunological evaluation was carried out in all the studies examined; one study evaluated the antioxidant status (TAS), total oxidant status (TOS), nitric oxide (NO), 8-hydroxy-2′-deoxyguanosine (8-Ohdg), myeloperoxidase (MPO), glutathione (GSH), malondialdehyde (MDA), and transforming growth factor-beta (TGF-β) and one study evaluated the MMP-1, MMP-8 and MMP-9 levels.

#### 3.2.3. NSPT and Airpolishing

Methods. The 3 studies selected for the review were randomized clinical trials published in English. The duration of the studies varied from 1 to 6 months: 1 study had a follow-up of 1 month, 1 study had a follow-up of 3 months and the last one a follow-up of 6 months; two studies had a too short treatment period for settlement of clinical improvement acts. The three clinical trials were conducted in Korea, Turkey and China. Randomization of studies and was performed with different methods: computer-randomized or tossing a coin.

Participants. Studies have on average recruited 36 patients, where 66.7% (two studies) included ≤36 patients and 33.3% (one study only) included >36 patients. The main inclusion criteria included: age ≥18 (100% of the studies examined), presence of at least a 2 teeth with at least 1 site with PPD and bleeding on probing in each quadrant, have not undergone periodontal surgery or nonsurgical periodontal treatment within 3 months prior to the start of clinical trials, without any systemic pathology that could affect periodontal clinical parameters and patients who did not require antibiotic prophylaxis for dental treatments.

Interventions. Patients underwent nonsurgical periodontal treatment (scaling and root planning in full-mouth or in quadrants) in addition to airpolishing with glycine and/or erythritol immediately after treatment (two studies with glycine and one study with erythritol).

Primary outcomes. Studies carried out evaluated PPD, CAL, BOP; all studies evaluated each type of adverse event. The frequency of evaluation of outcomes was variable: monthly or quarterly.

Secondary and additional outcomes. PI, GI, GR are clinical parameters evaluated in addition to the probing depth and the loss of clinical attachment; they have not been recorded in all the studies examined. Two studies performed a microbiological or immunological analysis; one study only evaluated the patient’s halitosis through volatile sulfur compounds (VSCs).

#### 3.2.4. NSPT and Probiotics

Methods. The 15 studies selected for the review were randomized clinical trials in English. The duration of studies varied from 1 to 12 months for a total average of 5 months, where: 26.7% of studies had a follow-up of 1 month, 6.7% a follow-up of 2 months, 26.7% a follow-up of 3 months, 6.7% a follow-up of 9 months and 13.3% a follow-up of 1 year. In order to establish the effectiveness of therapy, it is necessary to ensure a follow-up of 3 months: so, the 33.4% of studies analyzed are insufficient to assess improvements. The portion of 26.7% of the studies were conducted in India, 20% in Turkey, 14.3% in Chile and 40% in China, Brazil, Iran, Spain, Pakistan, Arabia (equally distributed). Randomization of studies was performed using different methods: computer-based randomization and sealed opaque envelopes; one study was designed in split-mouth (for scaling and root planning (SRP) treatment).

Participants. Studies on average recruited 42 patients (42.86), where 80% of studies included ≤46 patients and 20% included >46 patients. The main inclusion criteria included: age ≥18 (100% of the studies examined), presence of at least a 2 teeth with at least 1 site with PPD and bleeding on probing in each quadrant, have not undergone periodontal surgery or nonsurgical periodontal treatment within 3 months prior to the start of clinical trials, without any systemic pathology that could affect periodontal clinical parameters and patients who did not require antibiotic prophylaxis for dental treatments.

Interventions. Patients underwent nonsurgical periodontal treatment (scaling and root planning in full-mouth or in quadrants) in addition to the administration of probiotics, mainly bifidobacteria and lactobacilli. Some of the studies included used one or more test groups (four clinical trials, 26.7% of the total studies) and one or more control groups (one clinical trial, 6.7% of the total studies). In the majority of studies probiotics were administered in lozenges (in two studies probiotics were administered in mouthwash and in one study they were administered in sachets).

Primary outcomes. In most of the studies they were PPD and CAL; all studies evaluated each type of adverse event. The frequency of evaluation of outcomes was variable: monthly, quarterly, half-yearly or a single final evaluation at one year.

Secondary and additional outcomes. BOP, PI, SBI, GI, GBI, MGI, MBI are clinical parameters evaluated in addition to the probing depth and the loss of clinical attachment; they have not been recorded in all the studies examined. Some studies have also performed microbiological and immunological (60% of the total); one study has also evaluated halitosis through ORG and BANA tests.

#### 3.2.5. NSPT and Chlorhexidine

Methods. The 15 studies selected for the review were randomized clinical trials in English. The duration of studies varied from 1 to 12 months for a total average of 3 months (3.8), where: 20% had a follow-up of 1 month, 53.3% a follow-up of 3 months, 20% a follow-up of 6 months and 6.7% a follow-up of 1 year; 20% of studies, that have had a follow-up of 1 month, are unsuitable to provide effective results in pocket depth and in the gain of clinical attachment. 46.7% of the studies were conducted in India, 13.3% in Germany and 40% in Bosnia and Herzegovina, Iran, Italy, Arabia, Brazil, Spain (also distributed). Randomization of the studies was performed with different methods: computer-generated table, where the majority of allocation concealment was done through the use of opaque sealed envelopes, or tossing/flipping a coin; the 40% of clinical trials was designed in split-mouth.

Participants. Studies have on average recruited 36 patients (36.3), where 66.7% of studies included ≤36 patients and 33.3% included >36 participants. The main inclusion criteria included: age ≥18 (100% of the studies examined), presence of at least a 2 teeth with at least 1 site with PPD and bleeding on probing in each quadrant, have not undergone periodontal surgery or nonsurgical periodontal treatment within 3 months prior to the start of clinical trials, without any systemic pathology that could affect periodontal clinical parameters and patients who did not require antibiotic prophylaxis for dental treatments.

Interventions. Patients underwent nonsurgical periodontal treatment (scaling and root planning in full-mouth or in quadrants) in addition to rinsing or gingival irrigations with chlorhexidine after therapy (mouthwash, gel, chip or varnish). Some of the included studies used one or more test groups (60% of the total studies) and one or more control groups (13.3% of the total studies).

Primary outcomes. In most of the studies they were PPD and CAL; all studies evaluated each type of adverse event. The frequency of evaluation of outcomes was variable: monthly, quarterly, half-yearly or a single final evaluation at one year.

Secondary and additional outcomes. BOP, PI (PS), GR (REC), PBI, GI, BGI are clinical parameters evaluated in addition to the probing depth and loss of clinical attachment; they have not been recorded in all studies examined. Some studies have also performed microbiological (or analysis of plaque) and immunological (46.7% of the total); one study has also taken into analysis systemic and hematological parameters.

### 3.3. Synthesis of Results

#### 3.3.1. PPD

##### PPD and Laser

The probing depth has improved in all clinical trials examined; only one study did not consider PPD as a clinical parameter. For a total of 35 studies, therefore, although there was an improvement, there were no statistically significant differences between treatment groups in 65.7% of studies; there was a higher gain in pocket depth for the test group (statistically significant difference between treatment groups) in 31.4% of studies (one study showed a greater reduction in PPD in the control group, subject to only scaling and root planning). A portion of 54.5% of these were treated with PDT or Er:YAG laser (equally distributed) and 45.5% with PBM (photobiomodulation), diode laser or Nd:YAG laser, a combination of Er:YAG and Nd:YAG or a combination of PDT and PBM, (equally distributed); there was an improvement in pockets greater than or equal to 7 mm [25,56,57,58,59] and a reduction in sites with PPD greater than 4.5 mm [60]. In some clinical trials, although there were no significant differences between treatment groups, the probing depth was significantly reduced in the test group [25,56,57,58,59,60,61,62,63,64,65].

##### PPD and Ozone

The probing depth has improved in all the clinical trials examined. For a total of 5 studies, therefore, although found an improvement, there were no statistically significant differences between treatment groups; one study found an improvement in the PPD parameter in favor of ozone therapy, although there were no significant differences [30].

##### PPD and Airpolishing

The probing depth has improved in all the clinical trials examined. For a total of 3 studies, therefore, although there was an improvement, there were no statistically significant differences between treatment groups.

##### PPD and Probiotics

The probing depth has improved in all the clinical trials examined. For a total of 15 studies, although there was an improvement, there were no statistically significant differences between treatment groups in 60% of studies; there was more gain in pocket depth for the test group (statistically significant difference between treatment groups) in 40% of studies. Of these 66.7% were treated with strains of lactobacilli and 33.3% with strains of bifidobacteria or a combination of lactobacilli and bifidobacteria (equally distributed). In some clinical trials, although there were no significant differences between treatment groups, the depth of the survey was reduced more in the test group for moderate and deep periodontal pockets [66,67,68].

##### PPD and Chlorhexidine

The probing depth has improved in all the clinical trials examined. For a total of 15 studies, although there was an improvement, there were no statistically significant differences between treatment groups in 53.3% of studies; there was more gain in pocket depth for the test group (statistically significant difference between treatment groups) in 46.7% of studies. Of these 57.1% was treated with chlorhexidine chip or chlorhexidine gel (equally distributed) and 42.9% was treated with chlorhexidine mouthwash, chlorhexidine varnish or a full-mouth disinfection protocol (then gingival irrigation with chlorhexidine gel, tongue brushing with chlorhexidine gel and rinsing with chlorhexidine mouthwash), equally distributed.

#### 3.3.2. CAL

##### CAL and Laser

The loss of clinical attachment has improved in all the clinical trials examined; three studies have not considered CAL (RAL) as a clinical parameter. For a total of 33 studies, therefore, although there was an improvement, there were no statistically significant differences between treatment groups in 66.7% of studies; there was a higher gain in clinical attack for the test group (statistically significant difference between treatment groups) in 33.3% of studies. Of these 54.5% was treated with PDT, PBM, or a combination of Er:YAG and Nd:YAG (equally distributed) and 18.2% with Er:YAG, diode laser or a combination of PDT and PBM, equally distributed; an improvement was observed in pockets greater than 4.5 mm [58] and over 7 mm [25,58].

##### CAL and Ozone

The loss of clinical attachment is improved in all the clinical trials examined. For a total of 5 studies there was an improvement in follow-up, although, even in this case, there were no significant differences between treatment groups; one study found a greater gain in clinical attachment in favor of ozone therapy, without significant differences [30].

##### CAL and Airpolishing

The loss of clinical attachment is improved in all the clinical trials examined. For a total of 3 studies, there was therefore an improvement in follow-up, although, again, there were no significant differences between treatment groups.

##### CAL and Probiotics

The loss of clinical attachment has improved in all clinical trials examined. For a total of 15 studies, although there was an improvement, there were no statistically significant differences between treatment groups in 75% of studies; there was more gain in pocket depth for the test group (statistically significant difference between treatment groups) in 25% of studies. Of these 66.7% were treated with lactobacilli strains and only one study with bifidobacteria strains. In some clinical trials, although there were no significant differences between treatment groups, the gain in clinical attachment was best in the test group, for moderate or severe periodontal pockets [67].

##### CAL and Chlorhexidine

The loss of clinical attachment improved in all clinical trials examined; two studies did not consider CAL as a clinical parameter. For a total of 13 studies, although there was an improvement, there were no statistically significant differences between treatment groups in 69.2%; there was a higher gain in clinical attachment (statistically significant difference between treatment groups) in 23.1% of studies (one study found a greater improvement in CAL in the control group, subject to only scaling and root planning [69]).

#### 3.3.3. BOP

##### BOP and Laser

Bleeding on probing improved in all clinical trials examined; 11 studies did not consider BOP. For a total of 25 studies, therefore, although there was an improvement, there were no statistically significant differences between treatment groups in 60% of studies; there was a higher gain in terms of BOP for the test group (statistically significant difference between treatment groups) in 12% of studies. Two studies have had greater improvements in bleeding at the poll for control groups [70,71].

##### BOP and Ozone

Bleeding on probing improved in all clinical trials examined; only one study did not consider BOP as a clinical parameter. For a total of 4 studies, there was therefore an improvement in follow-up, but there were no significant differences between treatment groups: one study reported significant improvements for bleeding on probing with the use of ozone [30].

##### BOP and Airpolishing

Bleeding on probing has improved in all the clinical trials examined. For a total of four studies, there was an improvement in follow-up, but there were no significant differences between treatment groups.

##### BOP and Probiotics

Bleeding on probing has improved in all clinical trials examined. For a total of 15 studies, although there was an improvement, there were no statistically significant differences between treatment groups in 72.7% of studies; there was a higher gain in terms of BOP for the test group (statistically significant difference between treatment groups) in 27.3% of studies (all treated with lactobacilli strains).

##### BOP and Chlorhexidine

Bleeding on probing has improved in all clinical trials examined; six studies did not consider BOP as a clinical parameter. For a total of 9 studies, therefore, although there was an improvement, there were no statistically significant differences between treatment groups in 66.7% of studies; there was a significant improvement for the test group (statistically significant difference between treatment groups) in 33.3% of studies.

#### 3.3.4. PI

##### PI and Laser

The plaque index has improved in all the clinical trials examined; seven studies have not considered PI as a clinical parameter. For a total of 29 studies, therefore, although there was an improvement, there were no significant differences between treatment groups in 82.7% of studies; there was a greater improvement in the plaque index for the test group (statistically significant difference between treatment groups) in 13.8% of cases (one study had a better result in the control group [47]).

##### PI and Ozone

The plaque index has improved in all clinical trials examined; only one study did not consider PI as a clinical parameter. For a total of four studies, there was therefore an improvement in follow-up, but there were no significant differences between treatment groups.

##### PI and Airpolishing

The plaque index has improved in all clinical trials examined; only one study did not consider PI as a clinical parameter. For a total of two studies, there was therefore an improvement in follow-up, but there were no significant differences between treatment groups.

##### PI and Probiotics

The plaque index has improved in all clinical trials examined; only one study did not consider PI as a clinical parameter. For a total of 14 studies, although there was an improvement, there were no statistically significant differences between treatment groups in 71.4%; there was a greater improvement in plaque indices for the test group in 28.6% of studies (all treated with lactobacilli strains).

##### PI and Chlorhexidine

The plaque index has improved in all the clinical trials examined; two studies have not considered PI as a clinical parameter. For a total of 13 studies, although there was an improvement, there were no statistically significant differences between treatment groups in 46.1% of studies; there was a greater improvement in the plaque index for the test group (statistically significant difference between treatment groups) in 46.1% of studies (one study achieved a better result in SRP treatment [69]).

#### 3.3.5. Microbiological and Immunological Analysis

##### Microbiological and Immunological Analysis: Laser

Improvements were found in bacterial count [72], in the number of pathogens belonging to the red complex [59] and orange [59,73], a reduction of aggregatibacter actinomycetemcomitans, porphyromonas gingivalis, prevotella intermedia, prevotella nigrescens, tannerella forsythia [24], an improvement in levels IL 1-β and TNα [26] and ratio levels of IL1-β and IL-10 [59].

##### Microbiological and Immunological Analysis: Ozone

There were improvements in GCF PTX-3 levels [29], an increase in TGF-β levels [73,74], a significant reduction in the number of bacteria present in subgingival plaque [30], especially prevotella intermedia [75]; one study found increased MMP levels in patients with chronic periodontitis and decreased MMP levels in patients with aggressive periodontitis [76].

##### Microbiological and Immunological Analysis: Airpolishing

There was an improvement in the number of bacteria present, with a difference in the relative expression of porphyromonas gingivalis [77] and a significant reduction in a quarterly follow-up of GCF [78].

##### Microbiological and Immunological Analysis: Probiotics

There have been improvements in bacterial count for obligatory anerobics [79], a reduction in red and orange complex bacteria [36,80], a reduction in proinflammatory cytokines [36] and an increase in TIMP-1 levels [81].

##### Microbiological and Immunological Analysis: Chlorhexidine

Improvements were found in anaerobic bacterial count [82], a reduction of the bacteria belonging to the red complex [83] and a reduction of capnocytophaga ssp. [84].

#### 3.3.6. Results of Single Studies and Bias

Appendix A (these tables reported the number of participants who completed the follow-up and were included in the analysis of the results).

Table 1 shows the risk of bias of the main articles examined. This review present a relatively low risk of bias.

## 4. Discussion

The purpose of this review was to assess the effectiveness of minimally invasive therapies, as adjuvants to nonsurgical periodontal therapy, in order to formulate operational protocols aimed at restoring optimal conditions, respecting biological tissues, but also to the reduction of bacteremia, in an increasingly evolving historical context.

The reduction of bacteriemia is the main objective of nonsurgical periodontal therapy: this would prevent the spread of proinflammatory mediators, such as cytokines, and increase systemic inflammation; in addition, trying to reduce the bacterial load of patients, would reduce the production of highly contaminating aerosols. We must promote the use of rinse chlorhexidine or hydrogen peroxide before each dental procedure, the adoption of new therapies such as laser and ozone, the administration of probiotics, in order to minimize the use of instruments that produce aerosols, such as ultrasound and air and periopolishing systems.

### 4.1. Rinse Pre-Treatment

As already mentioned, it is generally believed that a preoperative antimicrobial mouthwash reduces the bacterial charge inside the oral cavity. However, as indicated by The Guidelines for the diagnosis and treatment of new coronavirus pneumonia (fifth edition), chlorhexidine is not effective to kill SARS-CoV-2 [2]. It is vulnerable to oxidation, therefore, the use of oxidizing agents containing 1% hydrogen peroxide or 0.2% iodopovidone is recommended in order to reduce the salivary load of oral microbes, including the potential transport of virus [85,86], before all dental procedures; a mouthwash based on ozonated olive oil or the use of ozonated water appliances could be useful in this respect. Ozone, in fact, seems to be effective against virus [87].

### 4.2. Modified Full-Mouth Disinfection and Chlorhexidine

Chlorhexidine has always been used in nonsurgical periodontal treatment due to its antimicrobial, bactericidal and bacteriostatic effect (in concentrations from 0.12% to 0.20%), high substantivity and lack of systemic toxicity [37], effective in the reduction of periodontal indexes [88,89], resulting in this way an antiseptic useful to produce benefits in patients with periodontal disease [69,82,83,84,88,89,90,91,92,93]; other authors, instead, argue that the use of chlorhexidine is not able to produce additional benefits to the treatment, compared to only scaling and root planning [94], having a minor role even in the long term [95].

In 1995, Quirynen introduced the “full mouth disinfection” treatment to prevent reinfection of treated sites during conventional periodontal therapy (quadrant therapy). The treatment consists of scaling and root planning performed over a period of 24 h, combined with gingival irrigation with 1% chlorhexidine gel, tongue brushing with 1% chlorhexidine gel and final rinsing with 0.2% chlorhexidine mouthwash. Then the patients follow a 2-month home protocol, continuing to brush the tongue with a 1% chlorhexidine gel and a rinse with 0.20% mouthwash, twice a day. This approach is effective in reducing microbial load, resulting in improved periodontal indices [96].

In 2014, a study conducted by the Tuscan Stomatological Institute, proposed an amendment to the Quirynen protocol, elaborating the concept of modified full-mouth disinfection, with the idea that it was more effective to get the reduction of bacterial load before the session of scaling and root planning, performing the disinfection protocol with chlorhexidine two weeks before treatment. So the first session is dedicated exclusively to the motivation and instruction of the patient in order to reduce bacteremia and recondition tissues, establishing a relationship of trust: this approach aims to reduce the bacterial load and clinical indices, thus reducing the patient’s discomfort, pain and bleeding before they can proceed to the treatment itself. Then we proceed to the actual treatment in full-mouth, with consequent revaluation in order to establish a tailor-made periodontal support therapy [97]. Therefore the results of this study showed a significant reduction of bleeding, thus favoring both the patient’s comfort and the operator’s procedure. In addition, the classic full-mouth approach highlights that the experience perceived by the patient is a fundamental component of the overall effect in nonsurgical full-mouth periodontal therapy, with anesthesia. In fact, the negative experience of anxiety results in a greater evasion resulting in delay in oral hygiene appointments and worsening of oral health [98].

The modified full-mouth, on the other hand, supports the modern trend towards more patient-centered approaches that become a proactive part of preventive therapy. Therefore, the real advantage of the two-week preparation period is the patient’s understanding of the therapy, which, acting beforehand on the clinical symptoms before treatment, allows the use of local anesthetics [97,98].

### 4.3. Measures for Modified Full-Mouth Disinfection

We have already seen that in this case it is the risk of pre-therapy bacteremia, which allows to avoid manual overtreatment and tissue contraction, reducing the discomfort for the patient. Furthermore, in order to reduce the contaminating aerosol, in this case, an ultrasonic ablator can be used in ‘soft-mode’ mode, thus reducing the width of movement of the insert, maintaining the same frequency (Figure 3).

However, as indicated by the Guidelines for the diagnosis and treatment of new coronavirus pneumonia (fifth edition), chlorhexidine, is not effective to kill SARS-CoV-2. It is vulnerable to oxidation, so it is recommended to use oxidizing agents containing 1% hydrogen peroxide or 0.2% iodopovidone, in order to reduce the salivary load of oral microbes, including the potential transport of virus [2]. So, in the light of this statement, we might consider modifying the pre- patient’s home protocol treatment, for example the use of a gel based on sunflower oil ozonized 15% and rinse with a mouthwash based on ozonated olive oil.

### 4.4. Laser Therapy

For many years, the laser has been recommended as an additional or additional protocol in the treatment of nonsurgical periodontal disease [73], due to its ability to achieve tissue ablation effects, hemostats, bactericides and detoxifiers against periodontal pathogenic bacteria [99].

Several studies have proven the effectiveness of laser use during periodontal therapy, highlighting an improvement in depth of survey, in the gain of clinical attachment and in the reduction of gingival bleeding in moderate and severe pockets: after a single application of the 810 nm diode laser, improvements have been achieved in parameters such as PPD and CAL [56,73,100] but also in levels of IL1-β [62]; other authors, however, argue that there are no further benefits in clinical parameters with the use of a diode laser, compared to/e solo/e scaling and root planning [101]. Also photodynamic therapy produces benefits in periodontal pockets [102], in the index of gingival bleeding and in gingival inflammation [64], as well as being effective in single-rooted teeth with aggressive periodontal disease [59], as well as the PBM [61,103,104,105] and Er:YAG laser [23,58,106], which is also favourable for the reduction and control of the proliferation of microorganisms [24,70], and Er,Cr:YSGG [63,107]. Less significant, however, is the Nd:YAG laser if not used in combination with Er.YAG laser, thus improving clinical and microbiological parameters for moderate and severe pockets, even in areas difficult to reach [25,72].

### 4.5. Measures for Laser Therapy in Modified Full-Mouth Disinfection

The modified full-mouth disinfection protocol can also be revised with the use of the laser. We recommend the use of diode laser during the scaling and root planning session for about 20–30 s in each periodontal pocket (810–980 nm, 1.5 W), favoring biostimulation, decontamination and cauterization of the tissue. Use it at the end of treatment in the II session (Figure 4).

### 4.6. Ozonetherapy

Ozone is an allotropic form of the oxygen molecule that occurs naturally in the Earth’s atmosphere: in medicine, a mixture of pure oxygen and pure ozone is used in the ratio between 95% and 99.95% and between 0.05% and 5% [108,109,110], respectively. It is applied as a powerful disinfectant, able to control bleeding and the cleaning of wounds in soft tissues, improving healing, with the increase of the supply of local oxygen [109,111]. In addition, at a high concentration, it kills bacteria very quickly and is a thousand times more powerful than other anti-bacterial agents [108].

The antibacterial action is related to the ability to react with double lipid bonds, thus leading to the lysis of the bacterial wall and the distortion of the content of bacterial cells; entering the cell promotes the oxidation of nucleic amino acids. It also inactivates viruses by spreading through the protein coating in the nucleus, causing damage to viral nucleic acid [112]. The use of dentistry provides only the topical application in the form of gas, water or oil, with a multitude of effects: anti-microbial, analgesic, oxygenating, anti-edema, immunomodulating [113].

It is also associated in the treatment of periodontal disease, as it affects the cellular and humoral immune system by stimulating the proliferation of immunocompetent cells and the synthesis of immunoglobulins. Biologically active substances, such as interleukins, leukotrienes and prostaglandins, which are useful in reducing inflammation and wound healing, are synthesized after the application of ozone [78]. In recent years, several clinical trials have been conducted to evaluate the effectiveness of ozone therapy as an adjunct to nonsurgical periodontal therapy. It has been noted that ozone both in gaseous and aqueous form reduces the growth of aggregatibacter actinomycentemcomintans, tannerella forsythensis, treponema denticola, porphyromonas gingivalis and prevotella intermedia [114]. Recently, the antimicrobial activity against specific periodontal pathogens of ozonated water has been shown: its use has shown a significant reduction in the number of bacteria present in the subgingival plaque [30].

### 4.7. Measures for Ozone Therapy in Modified Full-Mouth Disinfection

The modified full-mouth disinfection protocol can also be revised with the use of ozone, given the recent studies that highlight its potential effect against SARS-CoV-2 virus. It might be useful to associate ozone therapy (gas or water) to the first motivational session, promoting home products for a continuous contribution of the active ingredient; in fact, toothpastes, gels and mouthwashes are available. Subsequently for the causal therapy can be applied both at the beginning and at the end of treatment: 3–4 close applications are desirable in order to stimulate the healing of the tissue (Figure 5).

### 4.8. Airpolishing

The use of airpolishing with glycine is effective in the removal of biofilm above and below gingival and is increasingly used in maintenance periodontal therapy, reducing the formation of bacterial colonies in periodontal pockets [115]: FFP2 mask respirators are critical to protect dental hygienists and dentists and their importance it has been recognized [116]. In fact in periodontal pockets from 1 to 4 mm, the airpolishing with glycine, using a classic nozzle, is more effective in the removal of the subgingival biofilm than manual or ultrasonic instruments; in pockets from 5 to 9 mm, we recommend a perio tip [117], without causing periodontal tissue damage [118]. Another powder used is erythritol, introduced later, which is also suitable for the removal of biofilm, for the size of its particles, relatively small, and for the more stable chemical properties, compared to glycine [119]; it is also effective against some periodontal bacteria, such as Porphyromonas gingivalis [120] and Aggregatibacter actinomycentemcomintans [34].

It is a therapeutic choice common in clinical practice: a good compliance of the patient and an effective initial treatment, through the use of laser and ozone therapy to reduce the bacterial charge in the periodontal pockets, minimizes its use in necessary cases.

### 4.9. Measures for Airpolishing

The use of a perio tip, then of inserts created specifically for a deplaquing subgingival, having a jet confined within the periodontal pocket, can help to reduce the production of aerosol contaminant. Another important consideration, to avoid greater exposure to the clinical risk of contracting the virus SARS-CoV-2, is the choice of suitable powders: glycine and erythritol, being two very fine powders. They are not instruments of choice for a supragingival deplaquing or for the removal of extrinsic spots, because they would require more time to use; in this case, it is preferable to use powders based on calcium carbonate, whose spherical particles have a particle size ranging from 45 μm to 75 μm, less abrasive than sodium bicarbonate and therefore less aggressive on the tissues (Figure 6).

### 4.10. Home Hygiene Advice: The Probiotic Theme

Taking up the concept of oral microbiome, to reduce the chemical–pharmacological action and maintain a proper microbiological balance within the oral cavity, probiotics or bio-inspired products based on probiotics, such as toothpastes, mouthrinses or chewable gums can be recommended. The issue of probiotics begins to make its way even more in the dental field: according to the official definition of the World Health Organization probiotics are living organisms that, administered in adequate quantities, bring a benefit to the health of the host. Currently it has been shown that they are able to provide beneficial effects to the organism through the mechanism of stabilization of the microbial flora and modulation of the immune system of the host. Bacteria capable of exerting one or more of these effects are predominantly lactobacilli or bifidobacteria.

Recent studies have highlighted the usefulness that probiotics could have for the prevention or treatment of certain diseases of the oral cavity such as caries, gingivitis and periodontitis, that are associated with a variation in the composition of the microbial flora and the activity of bacterial species as well as the reaction of the host. Several studies have proven their effectiveness in reducing anaerobic bacterial load [79] and highly virulent bacteria belonging to the red and orange complex [36,67,80], helping tissue healing, in terms of depth of survey, loss of clinical attack, gingival bleeding and plaque index.

In view of these considerations, toothpastes containing selected and tindalized probiotics are available on the market, namely Bifidobacterium and Lactobacillus, which have an antimicrobial action that allows blockage of the growth of pathogenic bacteria, but above all they are able to bind to the toxins that are released by the latter, inhibiting their action. In addition, these toothpastes also have inside them the Mastic of Chios, known for its bacterial and anti-inflammatory properties, which acts as an immunostimulant. The chewable gums, on the other hand, mentioned above, contain microrepair, therefore microcrystals of hydroxyapatite, added with three billion of selected probiotics, that is Lactobacillus reuteri, Lactobacillus salivarius and Lactobacillus plantarum, vitamin C, vitamin D3, calcium and zinc. It is recommended to use for at least 20 min once a day, preferably after a proper cleansing, and continued for 10 days. In addition, previously, dental machines, mouthwashes and gels based on olive oil or ozonated sunflower seed oil were also mentioned as a continuation of the causal therapy for a continuous release of ozone at the tissue level.

## 5. Conclusions

To understand human health in its entirety and to intervene in a timely manner on the disease with precise and effective bio-inspired therapies that aim not only to solve the problem in the immediate, but also to reduce or even avoid relapse, we absolutely need to know the oral microbiome. In addition, knowing in depth the stability or variability of the human microbiota will allow us to better assess the health status of each patient in a holistic sense: to periodontics, an analysis of the oral microbiome will allow to associate, beyond reasonable doubt, the presence and severity of a possible periodontal disease. Given the link of the latter with a long series of different diseases, the collaboration between the dentist and other specialist doctors will ensure more timely, accurate and calibrated care for each patient. Therefore, analyzing the available scientific literature, we can evaluate ever more minimally invasive therapies, which aim at the reduction of bacteremia and the reduction of periodontal reference indices, with advantageous results for both the operator and the patient: laser, ozone, probiotics, glycine and erythritol appear to be a valuable support for nonsurgical periodontal treatment. This concept is even more important in a global pandemic situation, where the reduction of the bacterial load present in aerosols becomes a primary outcome, in order to manage the clinical risk in public or private health environments, without necessarily abandoning the promising instruments adopted so far. Bio-inspired systems have shown improvements in clinical parameters of greater relevance, such as depth of survey, loss of clinical attack, gingival bleeding and the index of plaque, accompanied by a reduction in bacterial load and proinflammatory cytokines, but these improvements have not been statistically significant. The risk of bias of the present investigation is relatively low, but it is not absent. Additionally, some studies present more than one aspect with uncertain or high risk. Therefore, future randomized clinical studies on the topic would be welcomed.

Further studies, aimed at assessing the effectiveness of the success of nonsurgical periodontal therapy, adjuvant by the use of laser and ozone therapy, airpolishing, probiotics and chlorhexidine, on the reduction of the risk caused by the new coronavirus are probably needed. This will promote a better management of health-workers’ risks of generating contaminated aerosols from patients who undergo nonsurgical periodontal therapy; limiting the chemical–pharmacological action to an initial period in order to manage and respect the biological tissues and avoiding, finally, an overtreatment.

## Figures and Tables

**Figure 1 jcm-09-03914-f001:**
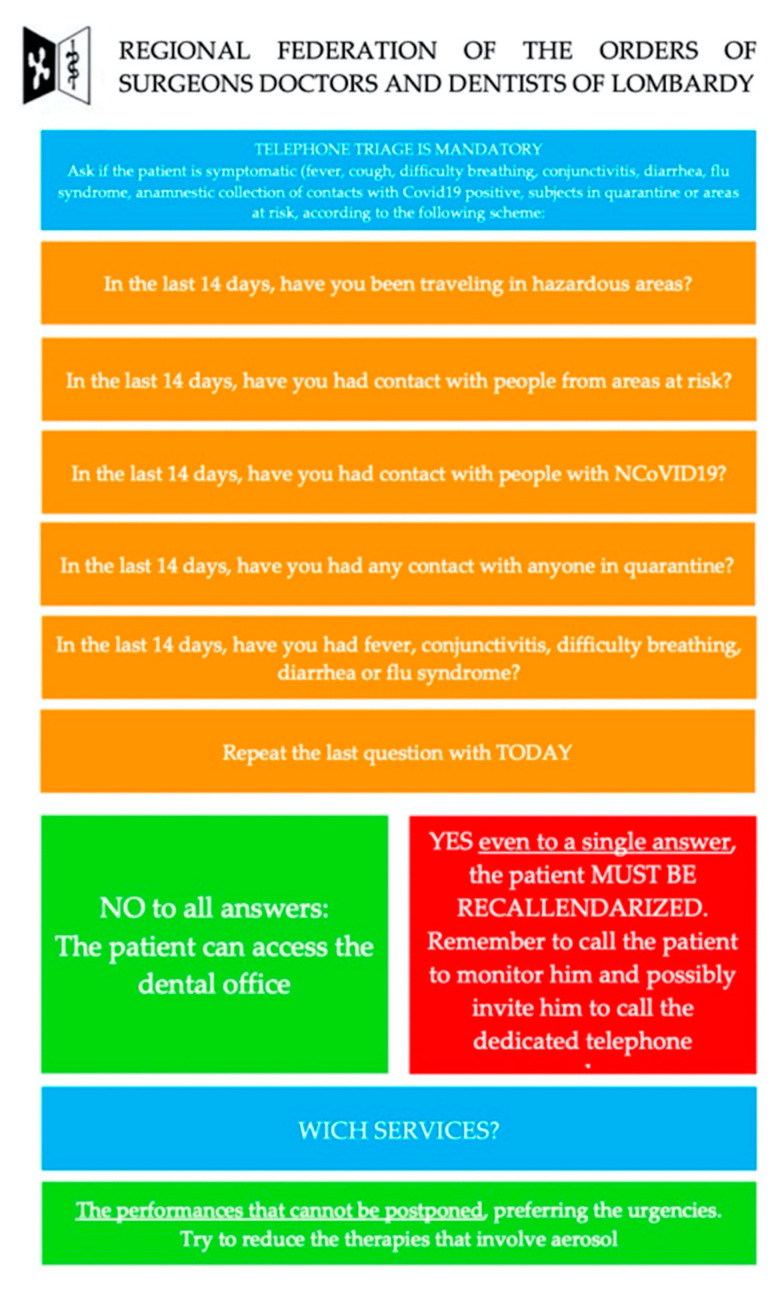
Telephone triage before entering the office.

**Figure 2 jcm-09-03914-f002:**
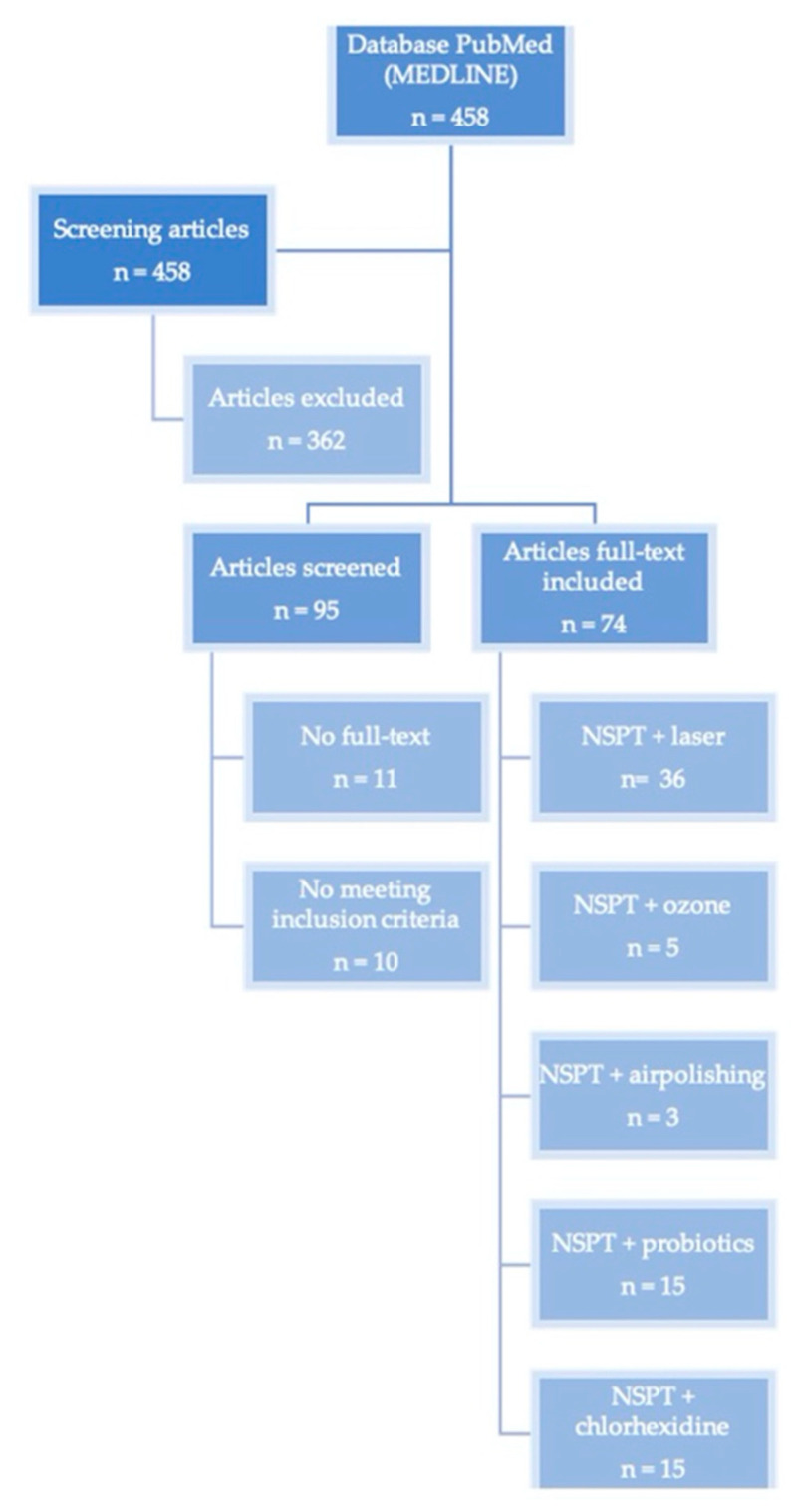
Articles included in the present investigation. NSPT: nonsurgical periodontal therapy.

**Figure 3 jcm-09-03914-f003:**
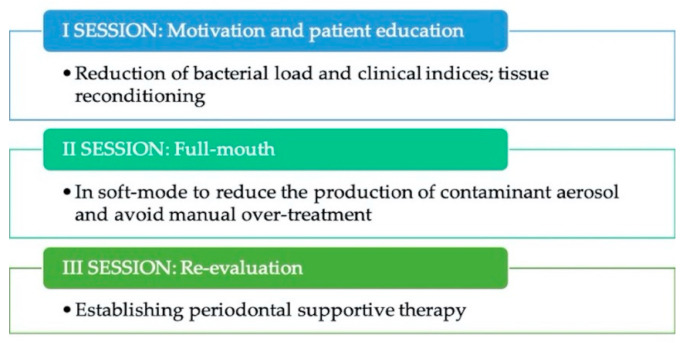
Sessions for modified full mout disinfection protocol.

**Figure 4 jcm-09-03914-f004:**
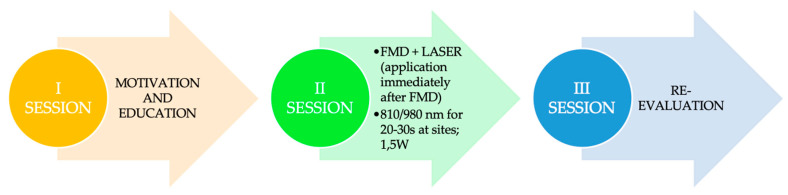
Laser as an adjuvant to the modified full mouth disinfection protocol. FMD: Full mouth disinfection; 1.5 W: 1.5 Watt.

**Figure 5 jcm-09-03914-f005:**
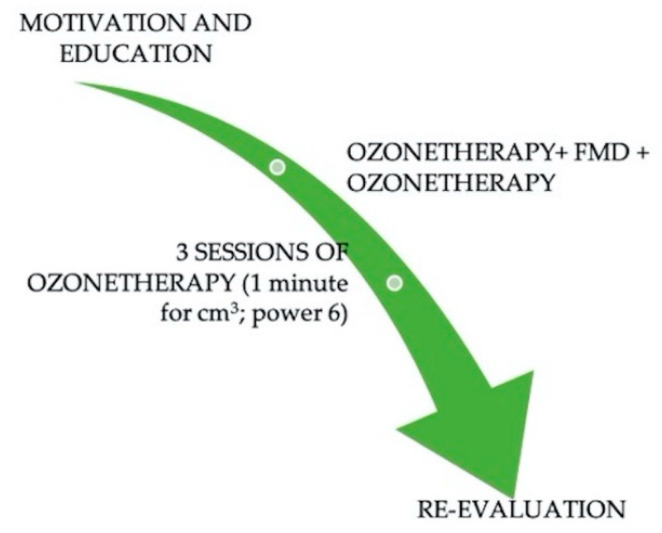
Ozone ad an adjuvant to the modified full mouth disinfection protocol. FMD: Full mouth disinfection.

**Figure 6 jcm-09-03914-f006:**
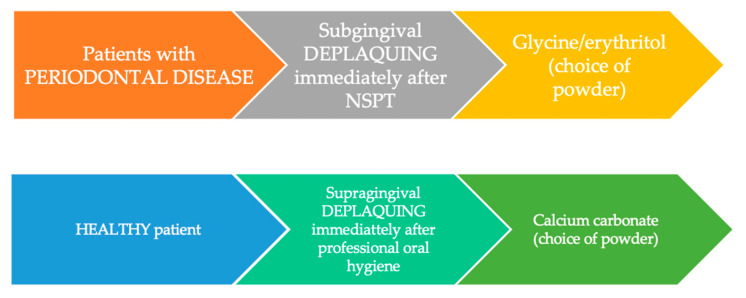
How to choose the type of powder, based on the oral health of the patient. NSPT: nonsurgical periodontal therapy.

**Table 1 jcm-09-03914-t001:** Risk of Bias.

	Adequate Sequence Generated	Allocation Concealment	Blinding	Incomplete Outcome Data	Registration Outcome
Dukić2013					
de Melo Soares2019	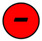		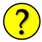		
Moreira2015					
Gündoğar2016					
Hayakumo2013	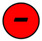		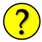		
Teughels2013					
Lecic2016	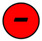	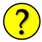	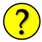		
Park2018	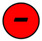	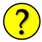	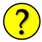		
Vivekananda2010	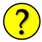				

Green symbol: Low risk of bias; Yellow symbol: Moderate risk of bias; Red symbol: Elevate risk of bias.

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
