# Peer review of "Bio-Inspired Systems in Nonsurgical Periodontal Therapy to Reduce Contaminated Aerosol during COVID-19: A Comprehensive and Bibliometric Review"

_jcm, 2020, doi:10.3390/jcm9123914_

Round 1
Reviewer 1 Report
The authors present an interesting and comprehensive review but with some points that should be reviewed.
The material and methods are very complete and very detailed, however, an explanation of how they assess the risk of bias is missing.
The results show a great work in terms of synthesis of the methodology and the results of the selected articles, but they should include an analysis of the risk of bias as I have commented previously.
The authors explain in results that most minimally invasive therapies improve the results of the variables studied, but nevertheless they tell us that in most of the studies the results were not significant. In fact, if most of them did not find significant differences, we cannot say that there was an improvement, so the final conclusion of the study does not conform to the reality found in the results. The authors state:
"Bioinspired minimally invasive systems are effective in improving the clinical parameters of greater relevance, such as depth of survey, loss of clinical attack, gingival bleeding and the index of plaque, accompanied by a reduction in bacterial load and proinflammatory cytokines." The reality is that in most studies the differences are not significant.
Author Response
Dear Reviewer,
Thank you for your kind words about our manuscript and for your suggestions. As requested, we tried to improve the accuracy of the text.
We have highlighted the modifications to the references with a green mark.
Please see the attachment.
The authors

Reviewer 2 Report
I am confused by your title--bacteremia reduction is a beneficial result of periodontal therapy, but how does that relate to the Covid-19 pandemic? Clinicians providing periodontal therapy are always concerned about becoming infected and affected from/by the patient's pathogens, and proper personal protective equipment is mandated during treatment. I suggest that you revise your manuscript to only include your good work in reviewing the various protocols that can reduce bacteremia and delete your editorial feelings about making that more important during the Covid pandemic than it would be in non-pandemic times. You've offered no evidence that bacteremia reduction has any benefit to the spread of Covid-19.
I have several specific comments as well:
line 23 your use of the phrase, 'minimally invasive therapies as adjuvants to non-surgical periodontal therapy,' in this context doesn't apply to your manuscript. All of your therapies include scaling and root planing, which is invasive. You could easily delete the minimally invasive term.
line 27 please define NSPT for the reader--a good place to do that would be in line 23.
line 78--you're narrow minded to think that a dental hygienist is the most exposed to bacteremia. A restorative procedure using a dental handpiece with water spray is at least equally risky.
line 127 There are several clinical trials using various 'surgical' diode wavelengths, not just those employing PDT. You have some of them as citations, so please include that category here.
line 205 LLT might be a typographical error. LLLT (Low Level Laser Therapy) is what you mean, although that acronym is not acceptable any longer. The correct current on is PBM--photobiomodulation.
line 229 you should indicate the months of the years that defined your search. Clearly, 2020 is not over yet.
line 235 you should include 'RCT' or equivalent in your listed search terms.
line 279 ff to 243 you're describing
line 283 studies with follow up periods of less than 3 months are considered unreliable to evaluate pocket depth reduction/clinical attachment gain due to the time necessary for re-attachment.
line 299 Your intervention need a bit more detail. You should briefly describe how scaling and root planing in addition to laser therapy was performed. In other words, does the laser therapy follow immediately after scaling or some other interval(s)? Lines 645 ff also do not indicate that--only 'during' the therapy. Figure 4 should be more specific in the green circle.
lines 358, 383, and 413--similar comment to line 299. (in Figure 5, you do offer a visual diagram of Ozone's place in your suggested protocol. It would be useful to have similar specific graphics for the others.)
line 586 you have not shown how reduction of bacteremia '...in an evolving historical content...' would affect the spread of Covid-19. Indeed, reduction of bacteremia is a noteworthy goal of any periodontal therapy anytime.
line 640 This is an important point--pre treatment rinse with peroxide or idoprovidone will reduce the transport of the Covid virus from the patient to the clinician. Then all dental procedures--NSPT, restorative, preventive, and prosthodontic--can then begin.
line 664 you do not describe how the diode can be used for full mouth disinfection. You are aware that the diode can unfavorable interact with pigmented accretions on the tooth surfaces.
lines 798-810 are exact duplicates of 630-642.
Author Response
Dear Reviewer,
thank you for your comments about our research. We have highlighted the modifications to the text with a yellow mark; there are some articles that have been added to the bibliography.
Here are the answers to your comments:
I am confused by your title--bacteremia reduction is a beneficial result of periodontal therapy, but how does that relate to the Covid-19 pandemic? Clinicians providing periodontal therapy are always concerned about becoming infected and affected from/by the patient's pathogens, and proper personal protective equipment is mandated during treatment. I suggest that you revise your manuscript to only include your good work in reviewing the various protocols that can reduce bacteremia and delete your editorial feelings about making that more important during the Covid pandemic than it would be in non-pandemic times. You've offered no evidence that bacteremia reduction has any benefit to the spread of Covid-19.
- It can therefore be said that most dental procedures produce doplets and contaminant aerosols: this risk is related the number of pathogens present in the aerosol/spray, and instruments, such as rotating instruments, ultrasonic scalers and piezo tools, produce a greater amount of spray and aerosols from other instruments such as air-to-water syringes. Bizzocca et al. attributing risk scores for the dental team and patients for each procedure to be: direct contact with saliva (score 1), direct contact with blood (score 2), production of low levels of spray/aerosol via air–water syringes (score 3), the production of high levels of spray/aerosol by use of rotating, ultrasound and piezoelectric tools (score 4): “tartar scaling” has one of the most high risk (7,5), like some surgical or endodontic procedures [13]. Dental team has always been used to counteract he risk of cross infections, by using face coverings, respirators, googles and face shields, gowns and coveralls in textile non-textile (TNT), gloves, disposables headgear and the covering of shoes: to reduce the risk of Covid-19 infection, those protective equipments become even more necessary, like CE-certified Filtering Face-Piece class 2 and class 3, in European Union[14].
- M. E. Bizzocca, G. Campisi, L. Lo Muzio, “An innovative risk-scoring system of dental procedures and safety protocols in the COVID-19 era”, BMC Oral Health 2020, vol. 20, no. 1, pp. 301-308.
- H. Xu, L. Zhong, J. Deng, et al., “High exspression of ACE2 receptor of 2019-nCoV on the epithelial cells of oral mucosa”, Int J Oral Sci 2020, vol.12, no. 1, pp.8.
- so the ACE2 expressing cells in oral tissues might provide possible routes of entry for the 2019-nCov, which indicate oral cavity might be a potential risk route of 2019-nCov infection
- Bad oral hygiene habits can encourage the accumulation of periodontal pathogens in the oral cavity and dysbiosis can accelerate the decline of lung function: in addition, pathogenic bacteria such as Treponema denticola, P. gingivalis, Fusobacterium nucleatum, Aggregatibacter actinomycetemcomitansand Veillonella parvula, were found in the lungs of patients admitted to the ICU [52]. Their presence, in addition, can not only change the microbial composition of the respiratory system, but also promote a number of responses of cytokines affect the immune homeostasis of the lungs: spherical levels of IL-6 and IL-8 increase significantly in patients with pulmonary dysfunction and local inflammatory factors spread into the systemic circulation. Changes in cytokines are assumed to reflect the state of the disease to a certain extent [53]. A high bacterial and viral load in the mouth can lead to complications in systemic diseases such as cardiovascular diseases, neurodegenerative diseases and autoimmune diseases , further supporting the bond between the mouth and the body: risk factors established for COVID-19 (age, sex and comorbidity) are also strongly implicated in imbalances in the oral microbiome. In fact, diabetes, blood hypertension and heart disease are associated with a greater number of F. nucleatum, P. intermedia and P. gingivalis, favoring the progression of periodontal disease: Patients with periodontal disease increase the risk for cardiovascular disease by 25%, for high blood pressure by 20% and triple the risk for diabetes mellitus [54,55,56]. Epithelial sensitization and haematogenic diffusion of proinflammatory mediators such as cytokines, produced in the periodontal diseased tissue, can increase systemic inflammation and decrease airflow: this can be exacerbated by the stimulation of the liver to produce acute phase proteins, such as interleukin-6, which boost the inflammatory response of the lungs and the rest of the body. Similarly, patients with COVID-19 in severe form also express systemic inflammation and higher levels of IL-6, IL-2, IL-10, TNF and C-reactive protein [58].
- S. Kumar, “From focal sepsis to periodontal medicine: a century of exploring the role of the oral microbiome in systemic disease”, J Physiol 2017, vol. 595, no. 2, pp. 465-476.
- Olsen, K. Yamazaki, “Can oral bacteria affect the microbiome of the gut?”, J Oral Microbiol 2019, vol. 11, pp. 1586422.
- Takahashi, f. Nishimura, M. Kurihara, et al., “Subgingival microflora and antibody responses against periodontal bacteria of young Japanese patients with type 1 diabetes mellitus”, J Int Acad Periodontol2001, vol. 3, no. 4, pp. 104-11.
- L. M. Paizan, J. F. Vilela-Martin, “Is there an Association between Periodontitis and Hypertension?”, Curr Cardiol Rev2014, vol. 10, no. 4, pp. 355-361.
- M. Presha, A. L. Alba, “Periodontitis and diabetes: a two-way relationship”, Diabetologia2012, vol. 55, no. 1, pp. 21-31.
- M. Aguiler, J. Suva, J. Buti, et al., “Periodontitis is associated with hypertension: a systematic review and meta-analysis”, Cardiovasc Res2020, vol. 116, no. 1, pp. 28-39.
- Sampson, N. Kamona, A. Sampson, “Could there be a link between oral hygiene and the severity of SARS-CoV-2 infections?”, Br Dent J. 2020, vol. 228, no. 12, pp. 971-975.
line 23 your use of the phrase, 'minimally invasive therapies as adjuvants to non-surgical periodontal therapy,' in this context doesn't apply to your manuscript. All of your therapies include scaling and root planing, which is invasive. You could easily delete the minimally invasive term.
- Bioinspired systems
line 27 please define NSPT for the reader--a good place to do that would be in line 23.
- NSPT (Non Surgical periodontal Therapy)
line 78--you're narrow minded to think that a dental hygienist is the most exposed to bacteremia. A restorative procedure using a dental handpiece with water spray is at least equally risky.
- Among the figures working in the dental team, one of the most exposed professional is definitely the dental hygienist
line 127 There are several clinical trials using various 'surgical' diode wavelengths, not just those employing PDT. You have some of them as citations, so please include that category here.
- From different clinical trials, in fact, it is clear that the main effects are found in the reduction of the probing depth and in the gain of the clinical attachment with the use of a Er:YAG laser (as described [26], after non surgical periodontal therapy [27], laser compared with SRP alone [28]), but also in bleeding on probing in case was used Nd:YAG laser, after non surgical periodontal therapy with ultrasonic and hand instruments[29,30]; further improvements were found in the decrease in levels of IL-1β and TNF-α following photodynamic therapy [31].
line 205 LLT might be a typographical error. LLLT (Low Level Laser Therapy) is what you mean, although that acronym is not acceptable any longer. The correct current on is PBM--photobiomodulation.
- PBM (photobiomodulation)
line 229 you should indicate the months of the years that defined your search. Clearly, 2020 is not over yet.
- January 2010 to March 2020
line 235 you should include 'RCT' or equivalent in your listed search terms.
- “RCTs AND laser AND periodontal disease”.
- “RCTs AND probiotics AND periodontal diasease”.
- ”,“RCTs AND glycine AND periodontal diasease”.
- “RCTs AND erythritol AND periodontal diasease”.
- “RCTs AND chlorexidine AND periodontal disease”
line 279 ff to 243 you're describing
line 283 studies with follow up periods of less than 3 months are considered unreliable to evaluate pocket depth reduction/clinical attachment gain due to the time necessary for re-attachment.
- these studies have some limits releated to pocket depth and clinical attachment gain, because the follow-up is too short to allow the re-attachment).
- Also in these clinical trials the follow-up is too short for evaluate improvements in clinical parameters.
- two studies have had too short treatment period for settlment of clinical improvement acts.
- In order to establish the effectiveness of therapy, it is necessary to ensure a follow-up of 3 months: so, the 33,4% of studies analyzed are insufficient to assess improvements.
- 20% of studies, that have had a follow-up of 1 month, are unsuitable to provide effective results in pocket depth and in the gain of clinical attachment.
line 299 Your intervention need a bit more detail. You should briefly describe how scaling and root planing in addition to laser therapy was performed. In other words, does the laser therapy follow immediately after scaling or some other interval(s)? Lines 645 ff also do not indicate that--only 'during' the therapy. Figure 4 should be more specific in the green circle.
- The patients have undergone to scaling and root planing with ultrasonic and hand instruments: the laser was used, in most cases, immediately after non-surgical periodontal treatment, except for some lasers (like Er:YAG) that were used before therapy.
lines 358, 383, and 413--similar comment to line 299. (in Figure 5, you do offer a visual diagram of Ozone's place in your suggested protocol. It would be useful to have similar specific graphics for the others.)
- The patients have undergone to scaling and root planing with ultrasonic and hand instruments: ozone was used immediately after periodontal treatment.
- The patients have undergone to scaling and root planing with ultrasonic and hand instruments: ozone was used immediately after periodontal treatment.
- in addition to airpolishing with glycine and/or erythritol immediately after treatment.
- with chlorhexidine after therapy
line 586 you have not shown how reduction of bacteremia '...in an evolving historical content...' would affect the spread of Covid-19. Indeed, reduction of bacteremia is a noteworthy goal of any periodontal therapy anytime.
- The reduction of bacteriemia is the main objective of non-surgical periodontal therapy: this would prevent the spread of proinflammatory mediators, such as cytokines, and increase systemic inflammation. In addition, the decrease in periodontal pathogens would prevent them from spreading to the lung tract
line 640 This is an important point--pre treatment rinse with peroxide or idoprovidone will reduce the transport of the Covid virus from the patient to the clinician. Then all dental procedures--NSPT, restorative, preventive, and prosthodontic--can then begin.
- 1% hydrogen peroxide or 0.2% iodopovidone is recommended in order to reduce the salivary load of oral microbes, including the potential transport of virus [81,82], before all dental procedures
line 664 you do not describe how the diode can be used for full mouth disinfection. You are aware that the diode can unfavorable interact with pigmented accretions on the tooth surfaces.
- The modified full-mouth disinfection protocol can also be revised with the use of the laser. We recommend the use of diode laser during the scaling and root planing session for about 20-30 seconds in each periodontal pocket (810-980 nm, 1,5W), favoring biostimulation, decontamination and cauterization of the tissue. Use it at the end of treatment in the II session.
lines 798-810 are exact duplicates of 630-642.
- These lines were delected: it was a mistake.
Thank you again for your appreciations and for your suggestions.
The Authors.

Round 2
Reviewer 1 Report
The authors have improved the manuscript with the questions that were required.
Author Response
Dear Reviewer,
We are pleased to have understood and satisfied your requests.
The Authors.
Reviewer 2 Report
I cannot further review this manuscript because the author refuses to accept my comments about deleting the Introduction referring to COVID-19. As I indicated, the manuscript is very useful in its broad discussion about various protocols to reduce bacteremia. The author states that the purpose of the review is to assess protocols to reduce bacteremia (lines 22-24) and that goal is very important with or without a pandemic. The new text, lines 30-34, is perfect and adequately mentions the pandemic. It's useful to mention the pre-pandemic PPE and how it's even more needed now (lines 88-92); however, lines 92-100 seem redundant, since wearing the aforementioned PPE is the protection, not to mention that this paper's focus is supposed to be reduction of bacteremia, a discussion of the virus's affinity for epithelial lung cells or stability on surfaces. If all these thoughts were much more condensed and simply stated, the paper would be much improved. An example is the statement in lines 209-212, and 220-222. However, a poor attempt to link the pandemic to the paper's topic is how ozone has not had demonstrated effectiveness to kill COVID-19 (lines 106-108.) The obvious question for me: then, why mention this? My main point continues to be that the author has failed to provide any evidence how the well discussed protocols would affect the pandemic, except for conjecture. The paper would be much better if the COVID-19 related text be deleted, as I suggested.
Author Response
Dear Reviewer,
We are sorry that we did not understand what was requested previously.
The aim of this work was precisely to evaluate the effectiveness of protocols for non-surgical periodontal therapy, in reducing bacteriemia.
The mention, or at least the parts of the text, of Covid-19 did not want to focus on the pandemic itself, but on how the primary goal of a therapy that aims to reduce the bacterial load, may also act on reducing aerosol production procedures.
We have highlighted the modifications to the references with a pink mark (in the attached file).
We hope that this will be enough: if it still considers some parts of the text inappropriate or inadequate, we will make a major and detailed change.
We tried to make it clear how important it is to minimize procedures that generate aerosols in this period.
Thank you for for your suggestions.
The Authors
